# Evidence of Nicotine Dependence in Adolescents Who Use Juul and Similar Pod Devices

**DOI:** 10.3390/ijerph16122135

**Published:** 2019-06-17

**Authors:** Rachel Boykan, Maciej L. Goniewicz, Catherine R. Messina

**Affiliations:** 1Renaissance School of Medicine, Stony Brook University, Stony Brook, NY 11794, USA; Catherine.Messina@stonybrookmedicine.edu; 2Roswell Park Comprehensive Cancer Center, Buffalo, NY 14203, USA; Maciej.goniewicz@RoswellPark.org

**Keywords:** e-cigarette, nicotine, dependence, tobacco

## Abstract

*Background*: The use of high-nicotine content e-cigarettes (so-called pods, such as Juul) among adolescents raises concerns about early onset of nicotine addiction. *Methods*: In this analysis of adolescents surveyed from April 2017–April 2018, we compare survey responses and urinary cotinine of pod vs. non-pod using past-week e-cigarette users aged 12–21. *Results*: More pod users categorized themselves as daily users compared to non-pod users (63.0% vs. 11.0%; *p* = 0.001); more pod than non-pod users had used e-cigarettes within the past day (76.2% vs. 29.6%; *p* = 0.001). More pod users responded affirmatively to nicotine dependence questions (21.4% vs. 7.1%; *p* = 0.04). Urinary cotinine levels were compared among those responding positively and negatively to dependence questions: those with positive responses had significantly higher urinary cotinine levels than those responding negatively. *Conclusions*: Adolescents who used pod products showed more signs of nicotine dependence than non-pod users. Pediatricians should be vigilant in identifying dependence symptoms in their patients who use e-cigarettes, particularly in those using pod devices.

## 1. Introduction

Electronic cigarettes (e-cigarettes) have soared in popularity to such an extent that the Surgeon General has declared their use an “epidemic”. Per the Centers for Disease Control and Prevention (CDC) report of 2018 National Youth Tobacco Survey data, one in five high school students is a current e-cigarette user [1]. One may associate this precipitous rise in popularity with the emergence of the fourth-generation e-cigarette products, pods. We previously reported that Juul and similar pod devices contain high concentrations of nicotine salts, and that urinary cotinine (primary nicotine metabolite) in adolescent pod users was higher than cotinine levels previously reported among adolescent smokers of conventional cigarettes [2]. We also reported that adolescent e-cigarette daily users were more likely to report using pods [3]. These findings raised important questions about the potential for early onset of nicotine dependence among adolescents who use pods. In this secondary analysis of a previously described cohort of adolescents [3], we compare use patterns of past-week pod vs. non-pod e-cigarette users in order to assess potential symptoms of nicotine dependence, and to correlate these outcomes with subjects’ measured urinary cotinine.

## 2. Materials and Methods

Between April 2017 and April 2018, 517 adolescents, ages 12–21, were recruited from three Stony Brook Children’s outpatient offices. All participants completed an anonymous survey regarding e-cigarette use and provided a urine sample. Urine from all e-cigarette users and a random sample of non-e-cigarette using controls was sent to Roswell Park Comprehensive Cancer Center for analysis of cotinine, the main metabolite of nicotine, and total 4-(methylnitrosamino)-1-(3-pyridyl)-1-butanol (NNAL), found only in users of combusted tobacco cigarettes. Urine samples with overly dilute (<10 mg/dL) or concentrated (>390 mg/dL) creatinine were excluded from analysis. The full analysis protocol has been described previously [3]. 

In this secondary analysis, past-week pod users (*n* = 21) were compared with past-week e-cigarette users who did not use pods (“non-pod users”) (*n* = 27). Subjects who used combusted tobacco were excluded. Urinary cotinine levels were evaluated and compared between the two groups (pod vs. non-pod e-cigarette users) in 42 participants who answered questions regarding nicotine dependence. Statistical analysis was conducted using Statistical Package for the Social Sciences (Version 25, IBM Corporation, Armonk, NY 10504, USA). Descriptive statistics were used to describe self-reported use patterns. The chi-square test of independence/analysis of variance were used to analyze relationships between self-reported use and urinary biomarkers. All tests of significance were two-tailed and significant at *p* < 0.05. This study was approved by the Stony Brook University Institutional Review Board (CORIHS#:2016-3912-F).

## 3. Results

Of the 517 participants in the full cohort, 14.3% (*n* = 74) had used e-cigarettes in the past week and 2.9% (*n* = 18) were past-week tobacco smokers. Pod users were younger than non-pod users: 60.0% (pod users) vs. 40.0% (non-pod users) were 12–14 years; 56.0% vs. 44.0%, 15–17 years; 22.2% vs. 77.8% were ages 18–21 (*p* = 0.06). Overall, the most common reasons for trying e-cigarettes were curiosity (67.3%), friends using them (57.1%), and flavoring (24.5%). There was no significant difference between the percent in each group that had ever tried a cigarette (45% of pod users vs. 33% of non-pod e-cigarette users; *p* = 0.324). There were no significant differences between the pod and non-pod groups with respect to gender, race, or ethnicity. 

Analysis of all survey results revealed that two-thirds (66.7%) of frequent (“use a lot”) past-week e-cigarette users used pods. Of “sometimes” users, 45.5% were pod users and 54.5% non-pod users. Of respondents who stated they had tried but no longer used e-cigarettes (so-called “experimenters”), 23.5% were pod and 68.5% non-pod users. More pod users categorized themselves as daily users compared to non-pod users (63.0% vs. 11.0%; *p* = 0.001); more pod than non-pod users reported having used an e-cigarette within the past day (76.2% vs. 29.6%, *p* = 0.001). 

We included five questions regarding nicotine dependence, for which any positive answer received a score of one, resulting in a total maximum score of 5 (Table 1). Of 48 past-week e-cigarette users with urine samples analyzed, 43.8% (*n* = 21) were pod users and 56.2% (*n* = 27) were non-pod e-cigarette users. Of these, 42 respondents answered questions about dependence, and 28.6% (*n* = 12) had at least one positive response. A higher percentage of these were pod (vs. non-pod) users (21.4%, *n* = 9 vs. 7.1%, *n* = 3; *p* = 0.04). Each of the three non-pod users who responded positively answered only one question affirmatively, giving each a total score of one. Of the pod users, five answered one question affirmatively, two answered two questions affirmatively, and one each answered four and five questions affirmatively. The cotinine concentration (mean ± SD) in urine samples from 12 positive respondents (nine pod users and three non-pod users) was higher than that of the 30 negative respondents’ cotinine: 675.40 ± 828.12 vs. 96.12 ± 298.31 ng/mL; *p* = 0.002. Six pod users vs. no non-pod users stated that they needed to vape upon awakening (*p* = 0.006); their cotinine levels were significantly higher than those of other respondents: 921.2 ± 960.4 vs. 148.2 ± 385.1 ng/mL (*p* = 0.001).

## 4. Discussion

Among adolescents who reported past-week e-cigarette use, pod users exhibited more signs of nicotine dependence with their more recent and more frequent use patterns than their non-pod using counterparts. Higher urinary cotinine levels were seen in those responding affirmatively to dependence questions. 

While stories of pod-addicted youth abound in the lay press, few studies have described or quantified adolescents’ symptoms of dependence on these products. In a study utilizing the Hooked on Nicotine Checklist (HONC), originally validated for use with smoking adolescents [4], McKelvey et al. described dependence symptoms in 34 adolescents and young adults who used e-cigarettes [5]. Similarly, Morean et al. implemented a four-item scale, the Patient-Reported Outcomes Measurement Information System (PROMIS) Nicotine Dependence Item Bank for electronic cigarettes PROMIS-E, originally designed for use with adults, to define dependence to e-cigarettes in a cohort of 520 adolescents [6]. 

In this study we utilized questions from the Autonomy over Tobacco Scale, the HONC, and the Fagerstrom Test for Nicotine Dependence [7,8]. While not validated for use together in this format, each question may be interpreted individually: any positive answer indicates a loss of autonomy over nicotine, a sign of nicotine dependence [9]. When all 10 HONC questions are used, the scores may be added to quantify the degree of dependence. Because we did not include all questions, our respondents’ total scores should not be taken as a true measure of relative dependence. 

Why should nicotine exposure and potential dependence be of such importance? Adolescents are particularly sensitive to the effects of nicotine, which may include attention and learning problems, anxiety, and depression [10,11,12]. Furthermore, while the long-term effects of these relatively new products have not been well established, to date, inhalation of e-cigarette aerosol has been associated with multiple concerning health effects, including endothelial dysfunction, toxicity from flavorings, volatile organic compound and ultrafine particle emissions, and increased risk for cardiovascular disease [11,13,14,15,16].

Our findings are limited by the small sample size. Additionally, while there was no significant difference between the percent of respondents in each group that had ever tried a cigarette, we did not ask how much or when past-week e-cigarette users who had been smokers had smoked in the past, so it is possible that some of them were nicotine dependent prior to using e-cigarettes, and therefore e-cigarette use could have served to maintain an addiction. However, given the decreasing prevalence of cigarette smoking among adolescents in the past several years [1], we would assume the number of teens transitioning from cigarettes to e-cigarette use would be small. In fact, numerous studies have shown that teens using e-cigarettes have increased odds of progressing to combusted tobacco use when compared to their non-e-cigarette using counterparts [17,18], but use of combusted tobacco has not been shown to lead to e-cigarette use [19]. 

The groups (pod users vs. non-pod users) may not have been mutually exclusive: pod users were defined as those who identified pod products when asked which e-cigarettes they used. It is therefore possible that the pod group used non-pod e-cigarettes in addition to pods. However, if anything, this would indicate that the cotinine level in a pod-only group might have been even higher. Similarly, non-pod e-cigarette users may have used but not reported using pods. We suspect, however, that it is unlikely that teens would have omitted to mention use of pods when asked, given their overall popularity. 

In this pilot study, pod users were more frequent users, and exhibited more signs of nicotine dependence. It would be logical to associate more frequent use with higher levels of dependence, but further study will be needed to quantify this relationship. It is also possible that teens using e-cigarettes in our study may have been more susceptible to nicotine dependence in general, and been more likely to use cigarettes, had cigarette use been more popular and available than e-cigarettes in their environment. Regardless, the use of nicotine in adolescents in either form is concerning. 

One might feel reassured that the majority of this past-week use group did not report symptoms of dependence. However, considering that 45% (9/20) of pod users indicated some loss of autonomy, and 30% (6/20) endorsed an even more ominous sign of dependence, needing to vape upon awakening, we feel the clinical implications of this small study cannot be ignored. 

While to date, no evidence-based treatments exist for nicotine-dependent youth, pediatricians may be able to help their patients by identifying dependence signs early. It is not enough to ask generally about vaping, or even ‘juuling’, but rather, pediatricians must focus on specific product use and behaviors such as daily or frequent use and vaping upon awakening. 

## 5. Conclusions

Adolescents who used pods showed more signs of nicotine dependence than those using non-pod products. Positive responses to dependence questions were reflected in higher urinary cotinine levels. Further study should elucidate whether nicotine dependence from e-cigarettes is characterized similarly to nicotine dependence from cigarettes in adolescents, and if use of higher nicotine-containing pods is associated with greater addictive symptoms than use of other e-cigarettes with lower nicotine concentrations. Given the potential adverse health consequences of nicotine use by adolescents, and the higher risk adolescent e-cigarette users have to progress to smoking, we advise that pediatricians should be vigilant in identifying symptoms of dependence in their patients who use e-cigarettes, particularly in those using pods.

## Figures and Tables

**Table 1 ijerph-16-02135-t001:** Dependence questions ** among 48 patients aged 12–21 who reported use of e-cigarettes in past week.

Statement	Total (%) Rated Agreement with Statement (*n* = 42)	Pod Users (%) Rated Agreement with Statement (*n* = 20)	Non-Pod Users (%) Rated Agreement with Statement (*n* = 22)	*p*
If I go too long without vaping, the desire to vape interrupts my thinking	3 (7)	3(15)	0 (0)	0.060
If I go too long without vaping, the desire to vape is so great that I need to vape again	2 (5)	2 (10)	0 (0)	0.130
If I go too long without vaping, I get angry or irritable	5 (12)	4 (20)	1 (5)	0.122
If I go too long without vaping, I get stressed	6 (14)	4 (20)	2 (9)	0.320
I need to vape when I awaken in the morning	6 (14)	6 (29)	0 (0)	0.006

** Not all respondents answered all questions.

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
