# Peer review of "Evidence of Nicotine Dependence in Adolescents Who Use Juul and Similar Pod Devices"

_ijerph, 2019, doi:10.3390/ijerph16122135_

Round 1

Reviewer 1 Report

This brief report examines whether users of "pod" vaping devices reported greater symptoms of dependence compared to other e-cigarette users. The authors report that users of pod devices were more likely to be daily users and to report symptoms of dependence, and had higher levels of urinary cotinine. Strengths of the paper include timeliness and importance given the widespread use of these products among younger populations.

Comments:

The authors refer to adolescents in the title and the abstract but also indicate that survey respondents were aged 12-21, so perhaps adolescent is not quite accurate.

While I appreciate that this is a brief report I think the authors have omitted some important details, including hypotheses.

Were the groups (pod vs non-pod) mutually exclusive?  From the description it appears that the latter excluded any pod use but the former may have included some non-pod use?

In the second paragraph of the Results there is reference to respondents who stated that they were no longer e-cig users.  These appear to be separate from the 48 participants described in the participants section, though it is not perfectly clear; however as the 48 are described as past week users these experimenters appear to be different. This is confusing and requires some revision.

If frequency of use is accounted for are there still meaningful differences in dependence self-report and cotinine?

It would also be useful to clarify the cotinine analyses.  In the Results section it appears that only samples from those who had positive responses to dependence questions were analyzed. If this is correct it should be reported more clearly previously (abstract, methods).

I appreciate that the long-term risks of chronic e-cig use are not well understood, but I think that the Discussion should include at least brief mention of potential implications. Why should pediatricians bother to be concerned about dependence via vaping?

Author Response

Dear Reviewer: 

We appreciate your thoughtful review of our paper. Below, please see responses to each of your points. 

The authors refer to adolescents in the title and the abstract but also indicate that survey respondents were aged 12-21, so perhaps adolescent is not quite accurate.

We have chosen to keep the title as it is, as the definition of adolescent in our practice, and as defined by the American Academy of Pediatrics is up until age 21, as noted in this policy statement: https://pediatrics.aappublications.org/content/140/3/e20172151.

While I appreciate that this is a brief report I think the authors have omitted some important details, including hypotheses.

We hope that further description of the larger study, added in this revision, addresses this point.

Were the groups (pod vs non-pod) mutually exclusive?  From the description it appears that the latter excluded any pod use but the former may have included some non-pod use?

We have addressed this in the limitations section, at the top of page 4

In the second paragraph of the Results there is reference to respondents who stated that they were no longer e-cig users.  These appear to be separate from the 48 participants described in the participants section, though it is not perfectly clear; however as the 48 are described as past week users these experimenters appear to be different. This is confusing and requires some revision.

We have added more detail about the overall study design, and have added some detail about this, in beginning of second paragraph of methods ("Analysis of all survey results revealed that...") 

If frequency of use is accounted for are there still meaningful differences in dependence self-report and cotinine?

We have included some discussion on p 4, lines 172 - 174. We feel that frequency of use is related to dependence, though these preliminary data are only suggestive. 

It would also be useful to clarify the cotinine analyses.  In the Results section it appears that only samples from those who had positive responses to dependence questions were analyzed. If this is correct it should be reported more clearly previously (abstract, methods).

We have added some clarification in both the abstract, lines 17 – 19, and the methods, lines 56-58.

I appreciate that the long-term risks of chronic e-cig use are not well understood, but I think that the Discussion should include at least brief mention of potential implications. Why should pediatricians bother to be concerned about dependence via vaping?

We have added a comment about this, p 3 lines 136 - 142.

Reviewer 2 Report

This study deals with a topic of high interest but thetopic is also very controversial. The paper is well written and easy to read.

My point of major critisism is on the recruitment of the 21 and 27 subjects. The recruiment needs to be mentioned in the paper and not just refered to as earlier described in reference 3. I am particularly interested in if these two groups are different already at the time of recruitment. Has e.g. the pod users a different nicotine history? Have they singled themselves out as beeing more and heavy smokers before recruitment and do they differ in other characteristics linked to tobacco/nicotine use and dependence. If that is the case there is an issue of possible reverse causality, i.e. the pos users are more dependent becasue they differed already at inclusion. If that would be the case the pod Juul would work better for them than the non-pod devices. The conclusions need to reflect this discussion.

The sample is small but that is what it is but it should be mentioned together with other limitations in the discussion.

Some more specific points.

Intro second line on the frequency of use. This issue is highly debated and just refereing to one study where one in five used e-cigarettes is not enough. Please add other views and figures.

Please also insert a sentence or two discussing the possibility that the users in this study would have been smoking cigarettes hadn´t the e-cigs been available. Adolescent smoking in US is declining despite e-cigs or is it becasue of e-cigs.

Author Response

Dear Reviewer: 

We appreciate your thoughtful review of our paper. Below, please see responses to each of your points. 

This study deals with a topic of high interest but thetopic is also very controversial. The paper is well written and easy to read.

Thank you!

My point of major critisism is on the recruitment of the 21 and 27 subjects. The recruiment needs to be mentioned in the paper and not just refered to as earlier described in reference 3. I am particularly interested in if these two groups are different already at the time of recruitment. Has e.g. the pod users a different nicotine history? Have they singled themselves out as beeing more and heavy smokers before recruitment and do they differ in other characteristics linked to tobacco/nicotine use and dependence. If that is the case there is an issue of possible reverse causality, i.e. the pos users are more dependent becasue they differed already at inclusion. If that would be the case the pod Juul would work better for them than the non-pod devices. The conclusions need to reflect this discussion.

We have added more description of the larger study and recruitment strategies, in the methods section and results, lines 65 and following, and in the discussion. In general, per NYTS, PATH and YRBS national data sets, the majority of adolescents who initiate e-cigarette use do not do so to quit smoking; the majority of them are nicotine-naïve. While we did not ask specific questions in our survey, we did not see a significant difference in the number of pod vs non-pod e-cigarette users who used cigarettes in the past. We discuss this further in the paper.

The sample is small but that is what it is but it should be mentioned together with other limitations in the discussion.

We have added this in discussion/limitations, line 143.

Some more specific points.

Intro second line on the frequency of use. This issue is highly debated and just refereing to one study where one in five used e-cigarettes is not enough. Please add other views and figures.

This data is from the CDC and NCI analysis of  NYTS data. To our knowledge these data are accepted and referred to, as the most accurate, by researchers in our field.https://www.cdc.gov/mmwr/volumes/68/wr/mm6806e1.htm

Please also insert a sentence or two discussing the possibility that the users in this study would have been smoking cigarettes hadn´t the e-cigs been available. Adolescent smoking in US is declining despite e-cigs or is it becasue of e-cigs.

We have added discussion about the progression to cigarettes from e-cigarette use. However, with respect, we do not feel that data support that adolescent smoking in US is declining because of e-cigarettes, but rather, because of significant public health measures that have been put into place in the last several decades, including clean indoor air laws, restrictions on advertising of cigarettes, high taxes, removal of flavors (except menthol) in 2009 under the Tobacco Control Act. We feel the rise of e-cigarettes is more likely from the lack of restrictions in this area, and the appeal to youth of flavors, sleek devices and easy accessibility. We have not elaborated on this in the paper, however, as we do not feel it is directly related to our topic. 

Round 2

Reviewer 2 Report

Thank you for an improved version of the manuscript.

However I insist on a point which you rejected. Let´s take an example from another area. If an adolescent start to use alcohol from soda type drink like cider you can often see a transition to other  and stronger alcoholic drinks. Is the softer alcohol laced drink a  gate-way or is its use just a sign of an interest in alcohol from the individual adolescent. Has the softer alcohol drink not existed would the adolescent not been interested in any stronger alcohol beverages at all? The same is true for nicotine/tobacco. One can not rule out that some adolescents interested in e-cig would have gone directly to cigarettes hadn´t the e-cig existed. Teens started with cigarettes before e-cigs were invented. This needs in all fairness to be acknowledged.

Author Response

Hello,

Thank you again for the review. In response to the comment below, we hope we have addressed that in the attached revision. The addition is highlighted in yellow, to distinguish it from the first track-changed edits. It can be found on page 4, lines 174- 177. 

We have also provided the response underneath the comment, below.

Sincerely,

Rachel Boykan

However I insist on a point which you rejected. Let´s take an example from another area. If an adolescent start to use alcohol from soda type drink like cider you can often see a transition to other  and stronger alcoholic drinks. Is the softer alcohol laced drink a  gate-way or is its use just a sign of an interest in alcohol from the individual adolescent. Has the softer alcohol drink not existed would the adolescent not been interested in any stronger alcohol beverages at all? The same is true for nicotine/tobacco. One can not rule out that some adolescents interested in e-cig would have gone directly to cigarettes hadn´t the e-cig existed. Teens started with cigarettes before e-cigs were invented. This needs in all fairness to be acknowledged.

It is also possible that teens using e-cigarettes in our study may have been more susceptible to nicotine dependence in general, and been more likely at to use cigarettes, had cigarette use been more popular and available than e-cigarettes in their environment. Regardless, the use of nicotine in adolescents in either form is concerning.